# The Hsp70-like StkA functions between T4P and Dif signaling proteins as a negative regulator of exopolysaccharide in *Myxococcus xanthus*

Pamela L. Moak, Wesley P. Black, Regina A. Wallace, Zhuo Li and Zhaomin Yang

Department of Biological Sciences, Virginia Polytechnic Institute and State University, Blacksburg, VA, USA

## ABSTRACT

*Myxococcus xanthus* displays a form of surface motility known as social (S) gliding. It is mediated by the type IV pilus (T4P) and requires the exopolysaccharide (EPS) to function. It is clear that T4P retraction powers S motility. EPS on a neighboring cell or deposited on a gliding surface is proposed to anchor the distal end of a pilus and trigger T4P retraction at its proximal end. Inversely, T4P has been shown to regulate EPS production upstream of the Dif signaling pathway. Here we describe the isolation of two Tn insertions at the *stk* locus which had been known to play roles in cellular cohesion and formation of cell groups. An insertion in *stkA* (MXAN_3474) was identified based on its ability to restore EPS to a *pilA* deletion mutant. The *stkA* encodes a DnaK or Hsp70 homolog and it is upstream of *stkB* (MXAN_3475) and *stkC* (MXAN_3476). A *stkB* insertion was identified in a separate genetic screen because it eliminated EPS production of an EPS$^+$ parental strain. Our results with in-frame deletions of these three *stk* genes indicated that the *stkA* mutant produced increased level of EPS while *stkB* and *stkC* mutants produced less EPS relative to the wild type. S motility and developmental aggregation were affected by deletions of *stkA* and *stkB* but only minimally by the deletion of *stkC*. Genetic epistasis indicated that StkA functions downstream of T4P but upstream of the Dif proteins as a negative regulator of EPS production in *M. xanthus*.

## INTRODUCTION

*Myxococcus xanthus*, a gram negative bacterium, exhibits complex social interactions during its life cycles (*Yang & Higgs, 2014*). When nutrients are plentiful, *M. xanthus* cells grow, divide and move over solid surfaces as social swarms in a vegetative growth cycle. Upon nutrient limitation, *M. xanthus* initiates a developmental cycle wherein cells aggregate on solid surfaces by their gliding motility. When these aggregates mature into multicellular fruiting bodies, rod-shaped vegetative cells within morph into spherical myxospores. These metabolically dormant myxospores can endure adverse environmental

Corresponding author
Zhaomin Yang, zmyang@vt.edu

elements such as heat, desiccation and UV radiation. When conditions become conducive for growth, myxospores germinate to reenter the vegetative cycle. Both developmental fruiting and vegetative swarming are multicellular behaviors which make *M. xanthus* a good model to study social or cell–cell interactions.

*M. xanthus* uses two genetically and morphologically distinct surface motility systems to facilitate its vegetative swarming and developmental aggregation (*Mauriello et al., 2010*). The adventurous (A) gliding system enables cells to move even when they are well isolated from one another. Social (S) gliding, on the other hand, only functions when cells are in close proximity or in groups. The bacterial type IV pilus (T4P) is known as the engine whose retraction powers *M. xanthus* S motility and bacterial twitching (*Mauriello et al., 2010*). Besides T4P, *M. xanthus* S motility requires the extracellular or exo-polysaccharide (EPS) to function (*Yang et al., 2014*). Available evidence supports a model wherein EPS on a neighboring cell or a surface triggers the T4P of *M. xanthus* to retract to actualize S motility (*Li et al., 2003*). This model explains why the function of S motility requires both T4P and EPS as well as cell proximity on most surfaces examined.

The T4P or Pil proteins as well as the Dif pathway play key roles in EPS regulation in *M. xanthus* (*Black, Xu & Yang, 2006*; *Yang et al., 2014*). *pilA* and other T4P⁻ *pil* mutants have been found to be EPS⁻. A *pilT* mutant, which is T4P⁺ with non-retractable T4P, is EPS⁺. Therefore, there is a positive correlation between the presence of T4P and EPS. Genes at the *dif* locus encode products related to bacterial chemotaxis proteins (*Yang et al., 1998*). DifA is homologous to the chemoreceptor MCP, DifC to the coupling protein CheW, and DifE to the histidine kinase CheA. Two additional proteins DifD and DifG are similar to the response regulator CheY and the phosphatase CheC, respectively. Null mutations in *difA*, *difC* and *difE* led to EPS⁻ and those in d*ifD* and *difG* to EPS overproduction (*Black & Yang, 2004*; *Yang et al., 2000*). Evidence indicated that DifE is a protein kinase and that it forms a ternary signaling complex with DifA and DifC (*Black et al., 2010*; *Yang & Li, 2005*). DifD and DifG influence the signaling strength of this DifACE complex by diverting phosphates from the kinase. Genetic studies showed that the T4P functions upstream of the Dif signaling proteins in EPS regulation (*Black, Xu & Yang, 2006*). The current model proposes that T4P function as physical sensors of other cells nearby. This sensory information is then relayed to the Dif pathway downstream to promote EPS production.

Of additional relevance to this work are the genes at the *stk* and the *che7* loci. A frameshift mutation in *cheW7* (*cheW7-1*) in the *che7* gene cluster (*Zusman et al., 2007*) restored EPS production to a *difA* deletion (Δ*difA*) mutant (*Black et al., 2009*). That is, a Δ*difA* single mutant is EPS⁻ but a Δ*difA cheW7-1* double mutant is EPS⁺. The Che7 chemosensory system likely plays an accessory role in EPS regulation in *M. xanthus* because a *cheW7* null mutation by itself does not impact EPS production in an otherwise wild-type (WT) background (*Black et al., 2009*). The *stk* locus had been identified previously because its mutations enhanced cellular cohesion in liquid culture and increased group formation at colony edges (*Dana & Shimkets, 1993*). Here we describe the isolation of two transposon insertions at the *stk* locus and the genetic characterization of *stkA* (MXAN_3474), *stkB* (MXAN_3475) and *stkC* (MXAN_3476). A *stkA* insertion was

found to suppress the EPS defect of a Δ*pilA* mutant whereas a *stkB* insertion was found to eliminate EPS production of a Δ*difA cheW7-1* strain. StkA is homologous to DnaK and HSP70 as described previously (*Weimer et al., 1998*). StkB shares similarity with the sterol carrier protein 2 (SCP2) or nonspecific lipid-transfer protein (NSLTP) (*Lev, 2010*; *Schroeder et al., 2007*). StkC is a small protein with limited homology to PhaE (*Goldman et al., 2006*), an enzyme involved in polyhydroxyalkanoate synthesis (*Han et al., 2010*). In-frame deletions were constructed for all three *stk* genes and their mutants were studied phenotypically. *stkB* and *stkC* deletions led to intermediate phenotypes in EPS production, motility and fruiting body development. Both *stkA* insertion and deletion restored EPS production to a Δ*pilA* mutant, but they failed to do so to a Δ*difA* strain. These results support a model wherein StkA functions downstream of T4P but upstream of the Dif pathway in the regulation of *M. xanthus* EPS production as a negative regulator. StkB and StkC are required for EPS production at the wild-type level and the absence of either reduced but did not eliminate EPS production.

## MATERIALS & METHODS

### Bacterial strains and growth conditions

*Escherichia coli* DH5α was used for plasmid constructions while DH5αλpir was used to clone transposon insertions from *M. xanthus* mutants. They were grown and maintained on Luria Bertani (LB) agar plates or in LB liquid medium (*Sambrook & Russell, 2001*). *M. xanthus* strains used in this study are listed in Table 1 and were grown and maintained on Casitone yeast extract (CYE) agar plates or in its liquid form (*Campos & Zusman, 1975*). Clone-fruiting (CF) agar plates were used to examine fruiting body development (*Hagen, Bretscher & Kaiser, 1978*). Plates for general use contained 1.5% agar. Soft agar plates, which were used to examine S motility, contained 0.4% agar (*Shi & Zusman, 1993*). Whenever necessary, kanamycin and oxytetracycline were supplemented to media at 100 μg/ml and 15 μg/ml, respectively (*Bellenger et al., 2002*; *Black & Yang, 2004*).

### Transposon mutagenesis and identification of transposon insertions

Transposon mutagenesis was performed using the *mariner*-based *magellan4* (*Rubin et al., 1999*) as previously described (*Black et al., 2009*). pMycoMar (containing *magellan4*) (Table 1) (*Rubin et al., 1999*) was electroporated into YZ101 (Δ*difA cheW7-1*) or DK10407 (Δ*pilA*). Cells were allowed to recover for 4 h and plated on CYE plates with Congo Red at 30 μg/ml. About 20,000 colonies were visually screened for EPS phenotypes after 5–7 days of incubation at 32 °C (*Black et al., 2009*).

The site of a Tn insertion in a mutant of interest was identified by cloning and DNA sequencing as has been described (*Black et al., 2009*). Briefly, genomic DNA from a mutant was digested with SacII (New England Biolabs) and allowed to self ligate. The ligation was transformed into DH5αλpir. Two primers, MarR1 and/or MarL1 (*Youderian et al., 2003*) were used to sequence the plasmids that were recovered from the transformant.

**Table 1 Strains and plasmids.** Plasmids and *M. xanthus* strains used in this study.

| Plasmid | Genotype or description | Source |
|---|---|---|
| pBJ113 | *M. xanthus* gene replacement vector | (*Julien, Kaiser & Garza, 2000*) |
| pBJ114 | *M. xanthus* gene replacement vector | (*Julien, Kaiser & Garza, 2000*) |
| pLZ407 | *stkA* in-frame deletion (Δ*stkA*) in pBJ113 | This study |
| pLZ429 | *stkB* in-frame deletion (Δ*stkB*) in pBJ114 | This study |
| pAM108 | *stkC* in-frame deletion (Δ*stkC*) in pBJ113 | This study |
| pMycoMar | *magellan4* mutagenesis vector | (*Rubin et al., 1999*) |
| **M. xanthus strain** | | |
| BY129 | *stkB1::Tn* in YZ101 | This study |
| BY801 | Δ *pilA stkA2::Tn* | This study |
| BY1129 | *stkB1::Tn* | This study |
| BY1801 | *stkA2::Tn* | This study |
| DK1622 | Wild type (WT) | (*Kaiser, 1979*) |
| DK10407 | Δ*pilA::Tet* | (*Wall & Kaiser, 1998*) |
| YZ101 | Δ*difA cheW7-1* (Δ*difA* suppressor strain) | (*Black et al., 2009*) |
| YZ601 | Δ*difA* | (*Xu et al., 2005*) |
| YZ603 | Δ*difE* | (*Black & Yang, 2004*) |
| YZ690 | Δ*pilA* | This study |
| YZ812 | Δ*stkA* | This study |
| YZ813 | Δ*stkB* | This study |
| YZ901 | Δ*pilA* Δ*stkA* | This study |
| YZ910 | Δ*stkC* | This study |
| YZ932 | Δ*difA* Δ*stkA* | This study |

## Construction of plasmids and strains

Plasmids used in this study are listed in Table 1. In-frame deletion alleles of *stk* genes were constructed using a two-step overlap PCR procedure (*Sambrook & Russell, 2001*). PCR products with the in-frame deletion alleles of *stkA* and *stkC* were blunt-end ligated into *Sma*I-digested pBJ113 (*Julien, Kaiser & Garza, 2000*) create pLZ407 and pAM108, respectively. The PCR product with the *stkB* in-frame deletion was similarly ligated into *Sma*I-restricted pBJ114 (*Julien, Kaiser & Garza, 2000*) to create pLZ429. The mutant alleles in pLZ407, pLZ429 and pAM108 deleted codons 5 to 535 for StkA, 5 to 108 for StkB and 5 to 85 for StkC, respectively.

A two-step procedure (*Ueki, Inouye & Inouye, 1996*) was performed to construct deletions of chromosomal *stk* genes. The three plasmids above were used to delete *stkA*, *stkB* and *stkC* from the WT strain (DK1622) to construct YZ812 (Δ*stkA*), YZ813 (Δ*stkB*) and YZ910 (Δ*stkC*) , respectively. In addition, pLZ407 was used to delete *stkA* from YZ690 (Δ*pilA*) and YZ601 (Δ*difA*) to create strains YZ901 (Δ*pilA* Δ*stkA*) and YZ932 (Δ*difA* Δ*stkA*).

## Examination of EPS production

EPS production was examined by two different assays: one qualitative and one quantitative. The qualitative assay utilized plates with 50 µg/ml calcofluor white (CW), a fluorescent dye that binds to EPS (*Black & Yang, 2004*; *Dana & Shimkets, 1993*). Cells from overnight cultures were pelleted and resuspended in MOPS (morpholinepropanesulfonic acid) buffer (10 mM MOPS [pH 7.6], 2 mM $MgSO_4$) at approximately $5 \times 10^9$ cells/ml. Then, 5 µl of the suspension were spotted onto CYE plates with CW and incubated at 32 °C for 6 days. Fluorescence under long-wavelength ($\sim$365 nm) UV illumination was directly photographed with a digital camera (*Black et al., 2009*). The binding of trypan blue was used to quantify EPS in a liquid assay (*Black & Yang, 2004*). Cultures grown overnight in CYE to $\sim 3.5 \times 10^8$ cells/ml were harvested, washed and re-suspended to approximately $2.8 \times 10^8$ cells/ml in MOPS buffer with 5 µg/ml trypan blue. The control samples contained trypan blue in MOPS buffer without cells. The samples were vortexed and incubated with shaking at 300 rpm at 25 °C for 30 min. The absorbance of the supernatants after centrifugation was measured at 585 nm. EPS production of all strains was normalized to that of the WT strain which was arbitrarily set as 1 (*Dana & Shimkets, 1993*). Quantitative experiments with trypan blue were repeated at least three times with each sample analyzed in triplicates and a representative data set is shown in the paper.

## Examination of motility and fruiting body development

Motility was examined by placing 5 µl of the cell suspension at $5 \times 10^9$ cells/ml onto the center of a standard (1.5% agar) or soft (0.4% agar) CYE plate. The standard agar plates were examined after 2 days and the soft agar plates after 5 days of incubation at 32 °C (*Black & Yang, 2004*; *Shi & Zusman, 1993*). For the examination of fruiting body formation, overnight cultures were harvested and resuspended in MOPS buffer at $5 \times 10^9$ cells/ml. Then, 5 µl of the suspension were spotted onto CF agar plates and development was observed after 5 days of incubation at 32 °C.

## RESULTS

### Isolation of two *M. xanthus* transposon mutants with altered EPS production

To identify genes involved in the regulation and/or production of EPS, two genetic screens were carried out to search for mutants with altered EPS levels. In the first screen, a *pilA* deletion ($\Delta pilA$) strain (DK10407), which is T4P$^-$ and EPS$^-$ (*Black, Xu & Yang, 2006*), was mutagenized by a transposon (Tn) and mutants were allowed to form colonies on agar plates with the dye Congo red (CR). *M. xanthus* EPS$^+$ colonies appear red and EPS$^-$ ones are yellowish orange on these plates (See 'Materials & Methods'). Among approximately 20,000 colonies screened, BY801 and BY802 were found to form red colonies, indicating that they contained suppressors of the *pilA* deletion. BY801 is discussed here and the work on BY802 has been published elsewhere (*Wallace et al., 2014*).

The suppressor phenotype of BY801 and its link to the Tn were confirmed by an alternative EPS assay and genetic linkage analysis, respectively. As shown in Fig. 1A, BY801
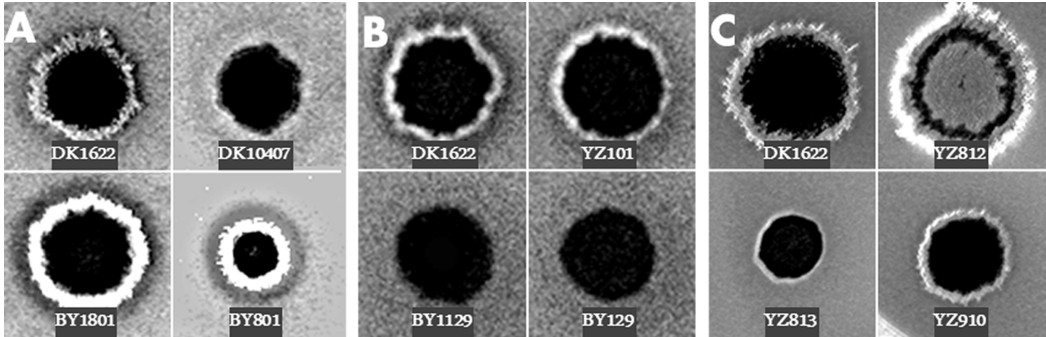

**Figure 1 EPS production of *stk* mutants.** Five microliter aliquots of cell suspension at $5 \times 10^9$ cells/ml of an indicated strain were spotted onto CYE plates with the fluorescent dye calcofluor white (CW) and florescence was documented under UV illumination after 6 days of incubation at 32 °C (See 'Materials & Methods'). (A) *stkA* insertion suppressed *pilA* deletion. (B) *stkB* insertion results in EPS defect. (C) Deletions of *stk* genes affect EPS levels to different extent. Strains: DK1622 (wild type), DK10407 (Δ*pilA*), BY801 (Δ*pilA stkA2::Tn*), BY1801 (*stkA2::Tn*), YZ101 (Δ*difA cheW7-1*), BY129 (Δ*difA cheW7-1 stkB1::Tn*), BY 1129 (*stkB1::Tn*), YZ812 (Δ*stkA*), YZ813 (Δ*stkB*), and YZ910 (Δ*stkC*).

was verified to be EPS+ as indicated by the fluorescence on a plate containing the dye Calcofluor white (CW). The Tn insertion in BY801 was transferred to the parental Δ*pilA* mutant by genomic DNA transformation (*Vlamakis, Kirby & Zusman, 2004*). Sixteen of the resulting transformants were examined and all were found to be EPS+. This established that a single Tn insertion locus in BY801 was responsible for Δ*pilA* suppression instead of any additional mutations elsewhere. When the Tn insertion was transferred to the WT background, the resulting strain BY1801 showed enhanced EPS production as indicated by increased CW binding in comparison with the wild type (WT) (Fig. 1A). These observations demonstrate that the Tn insertion in BY801 altered the function of a gene or genes important for *M. xanthus* EPS production and/or regulation. It should be noted that despite its ability to produce EPS, the colonies of BY801 differ from those of the WT (Fig. 1A) because the latter is S+ while BY801 is S− without *pilA*.

In the second genetic screen, the EPS+ mutant YZ101 (*cheW7-1* Δ*difA*) (*Black et al., 2009*) was used for Tn mutagenesis to identify additional genes critical for *M. xanthus* EPS production. About 70 EPS− mutants were identified from approximately 20,000 colonies on CR plates. Some of these mutants were reported previously (*Black et al., 2009*; *Lu et al., 2005*). BY129, which showed no obvious CR binding in the initial screen, is described here. Examination on plates with CW confirmed that BY129 has negligible EPS levels in comparison with its parent and the WT (Fig. 1B). When the insertion was re-introduced into YZ101 by genomic DNA transformation, the resulting transformants displayed the same EPS− phenotype as BY129 (not shown). When the Tn insertion was introduced into the WT strain, the resulting mutant BY1129 did not bind CW in plate assays (Fig. 1B). The gene(s) mutated by the Tn insertion in BY129 must play a role in EPS production and/or reguation in *M. xanthus*.

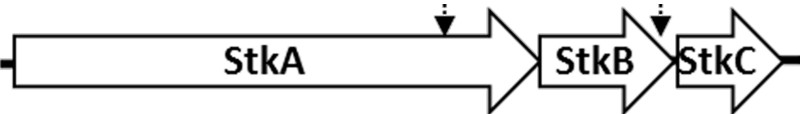

**Figure 2** *M. xanthus stk* **locus X.** The *stk* region shown is 2.54 kb with the ORFs of StkA, StkB and StkC indicated by open arrows approximately to scale. The inverted arrows indicate positions of Tn insertions in *stkA* and *stkB* in BY801 and BY129, respectively.

## Transposons in BY801 and BY129 inserted in two adjacent genes at the *stk* locus

The Tn insertions in BY801 and BY129 were identified as previously described (*Black et al., 2009*). In BY801, the insertion occured in MXAN_3474, a gene known as *stk* because of the *sti*cky phenotype of its mutant (*Dana & Shimkets, 1993*; *Kim, Ramaswamy & Downard, 1999*; *Weimer et al., 1998*) (Fig. 2). This gene will be designated as *stkA* and the insertion mutation here as *stkA2::Tn* hereafter (Table 1). *stkA* encodes a DnaK homologue of 540 amino acids (AAs) (*Goldman et al., 2006*) that was not found to be induced by heat shock (*Otani et al., 2001*). The *stkA2::Tn* insertion occurred in the 440th codon of *stkA* after a TA dinucleotide. In BY129, the Tn inserted into MXAN_3475, an open reading frame (ORF) of 141 codons 8 base pairs (bps) downstream of *stkA*. This ORF will be designated as StkB and the mutation as *stkB1::Tn* hereafter. StkB belongs to the superfamily of the sterol carrier protein 2 (SCP2) or nonspecific lipid-transfer protein (NSLTP) (*Lev, 2010*; *Schroeder et al., 2007*). Some members of this protein superfamily function in cholesterol trafficking and lipid metabolism as well as cell signaling in a variety of organisms (*Lev, 2010*; *Schroeder et al., 2007*). The *stkB1::Tn* insertion occurred after a TA dinucleotide in the 122nd codon. 12 bps downstream of *stkB* is MXAN_3476 (*Goldman et al., 2006*) or *stkC*. It encodes a protein of 89 AAs with limited homology to PhaE (*Goldman et al., 2006*), a polyhydroxyalkanoate synthetic enzyme (*Han et al., 2010*). The isolation of Tn insertions at the *stk* locus from independent genetic screens here and elsewhere (*Dana & Shimkets, 1993*) indicates that the *stk* genes are critical players in EPS production in *M. xanthus*.

## Δ*stkA* produces more EPS while Δ*stkB* and Δ*stkC* produce less

The transposons in BY801 and BY129 inserted at the 3′ ends of *stkA* and *stkB*, respectively (Fig. 2). As truncated StkA and StkB may retain part of their functions, these insertions could be leaky or even gain-of-function mutations. In addition, *stkA, stkB* and *stkC* may form an operon (Fig. 2) and both insertions could be polar on downstream genes. The previous *stkA* mutant harbors a Tn insertion as well (*Dana & Shimkets, 1993*). To clarify the roles of the *stk* genes in EPS production, in-frame deletions of these genes were constructed (See 'Material & Methods'). YZ812, YZ813 and YZ910 deleted *stkA*, *stkB* and *stkC*, respectively (Table 1). EPS production by these strains was examined by CW binding (Fig. 1C); the Δ*stkA* strain exhibited more whereas Δ*stkB* and Δ*stkC* exhibited less fluorescence than the WT in this assay. In addition, EPS levels of the *stk* deletion mutants were quantified by a liquid dye binding assay (*Black & Yang, 2004*). As show in Fig. 3, the Δ*stkA* mutant increased EPS production over the wild type by about 50%. The Δ*stkB*

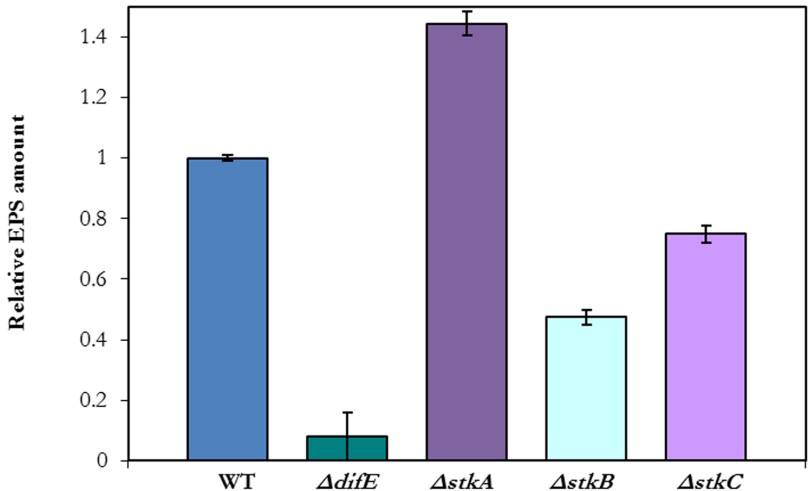

**Figure 3 The *stkA* mutant produced more EPS whereas the *stkB* and *stkC* mutants produced less.** EPS levels were quantified by a trypan blue binding assay (See 'Materials & Methods'). The amount of EPS for each strain was compared to that of the WT (DK1622) which was normalized to a value of one. Other strains are YZ812 (Δ*stkA*), YZ813 (Δ*stkB*), and YZ910 (Δ*stkC*) with the EPS⁻ strain YZ603 (Δ*difE*) as a control.

and Δ*stkC* mutants produced about 50% and 25% less than the WT, respectively. The results in Figs. 1C and 3 with the in-frame deletion mutants clearly implicate *stk* genes in *M. xanthus* EPS production. StkA is likely a negative regulator as its absence results in EPS overproduction. The roles of StkB and StkC are less clear, as the deletion of their genes led to intermediate EPS phenotypes. Phenotypic comparisons also indicate that the *stkB* insertion in BY129 and BY1129 is likely polar because it led to a more severe EPS defect than the deletion of either *stkB* or *stkC*. There are two additional ORFs (MXAN_3471 and MXAN_3472) upstream of and in the same orientation as StkA (*Goldman et al., 2006*). Their in-frame deletions resulted in no alteration in *M. xanthus* EPS production or any other phenotype examined (results not shown) and these two genes are not discussed in this manuscript.

## StkA functions downstream of PilA but upstream of Dif Proteins

EPS production is regulated in part by the Dif pathway (*Black, Xu & Yang, 2006*; *Yang et al., 2014*). T4P are proposed to perceive and relay signals downstream to Dif proteins to promote EPS production (*Black, Xu & Yang, 2006*). The relationship of StkA with Dif was examined by the construction of a Δ*difA* Δ*stkA* mutant. In addition, a Δ*pilA* Δ*stkA* mutant was constructed to confirm the suppression of Δ*pilA* by the *stkA* deletion. As shown in Fig. 4, the Δ*pilA* Δ*stkA* double mutant (YZ901) produced more EPS similarly as the Δ *stkA* single mutant (YZ812), indicating the suppression of Δ*pilA* by *stkA* null mutations. On the other hand, the Δ*difA* Δ*stkA* double mutant (YZ932) appeared similar to the Δ*difA* single mutant (YZ601) with both producing very little EPS (Fig. 4). The finding that the Δ*stkA* mutation is epistatic to a Δ*pilA* but not a Δ*difA* mutation led to the

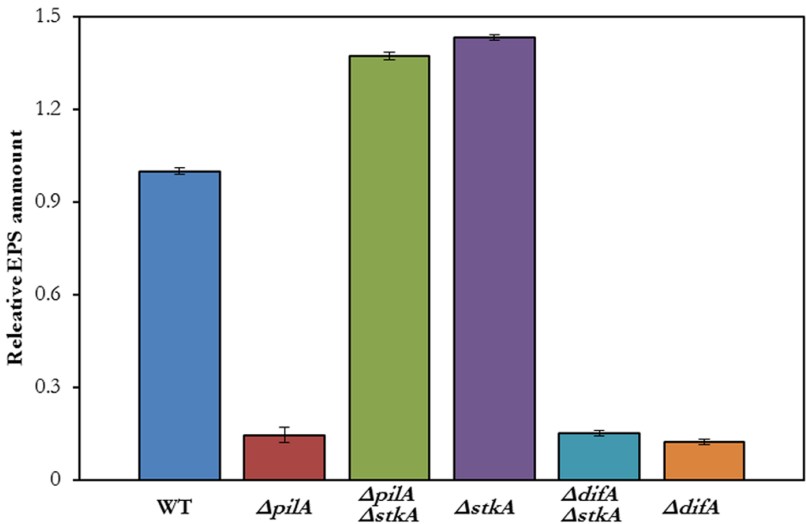

**Figure 4 ΔstkA suppresses ΔpilA but not ΔdifA.** EPS levels were quantified by a trypan blue binding assay as in Fig. 4. The strains are DK1622 (WT), YZ690 (ΔpilA), YZ901 (ΔpilA ΔstkA), YZ812 (ΔstkA), YZ932 (ΔdifA ΔstkA) and YZ601 (ΔdifA).

conclusion that StkA functions between T4P and the Dif pathway in the regulation of EPS production in *M. xanthus*.

### *stk* mutants show defects in motility

The surface motility of the *stk* deletions were examined first on regular agar plates (1.5% agar) which allow both A and S motility to contribute (*Shi & Zusman, 1993*) (Fig. 5A). Compared to the WT, the colony of the Δ*stkC* mutant (YZ910) appears only slightly smaller, consistent with the slight effect of the *stkC* deletion on EPS levels (Figs. 1C and 3). The colony morphology of the Δ*stkC* mutant was also highly similar to that of WT in its yellow pigmentation, high opacity as well as its rough surface and jagged edges. The Δ*stkA* and especially Δ*stkB* mutants showed a more diminished ability to spread on hard agar surfaces as their swarming colonies appeared smaller than that of the WT. The surface of the Δ*stkA* mutant colony is rougher and more elevated than that of the WT. The colony of Δ*stkB* is smoother, glossier and flatter than the WT. These observations are consistent with the Δ*stkA* mutant overproducing and the Δ*stkB* mutants significantly underproducing EPS in comparison with the WT.

Because EPS is essential for the T4P-mediated S motility, plates with 0.4% agar (soft agar plates) were used to examine S motility more specifically (*Shi & Zusman, 1993*) (Fig. 5B). The size of the swarming colony of the Δ*stkC* mutant is similar to that of the WT, indicating no obvious S-motility defect. The spreading of Δ*stkB* and Δ*stkA* mutants, especially the latter, was defective in comparison with the WT. Interestingly, except the size, the colony morphology of the Δ*stkA* mutant more closely resembled that of the WT. There were obvious swarming zones or rings for strains of the WT and the Δ*stkA*, but not for those of the Δ*stkB* and the Δ*stkC*. Overall, these results indicate that StkA and StkB

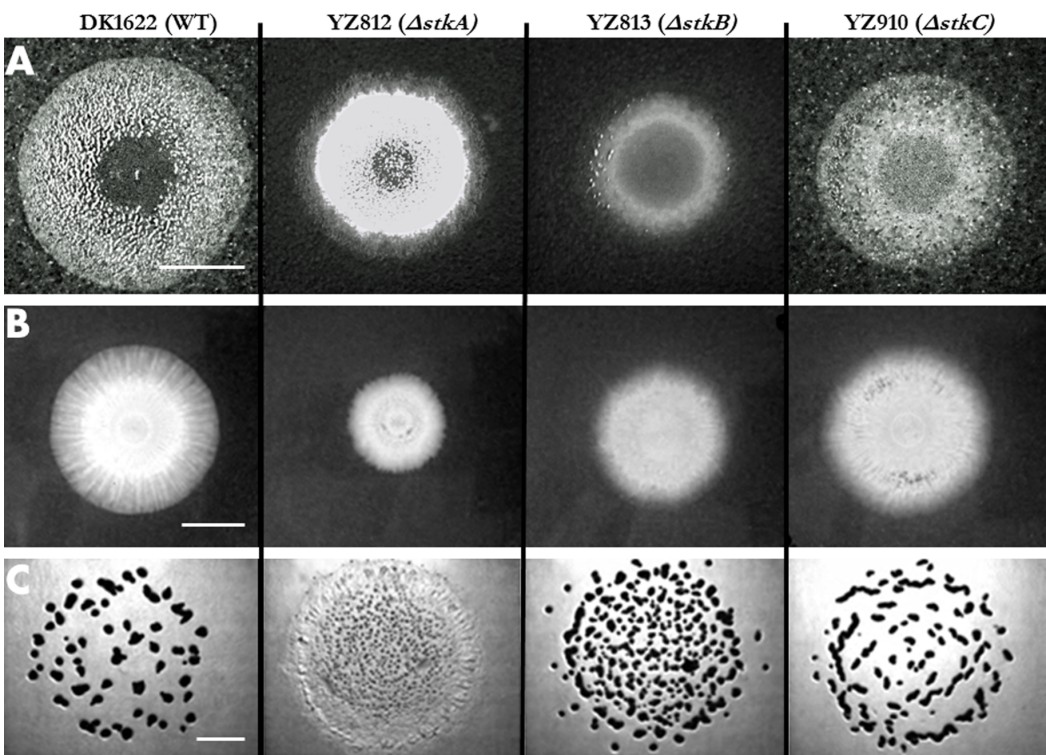

**Figure 5 Motility and developmental aggregation of *stk* mutants.** A 5 µl aliquot of the cell suspension at $5 \times 10^9$ cells/ml for each strain were plated in the center of a CYE plate with 1.5% (A) or 0.4% agar (B) to examine motility. The same amount of cells of a strain was spotted onto a CF plate to examine development (C). Results were documented after incubation at 32 °C (See Materials and Methods). The scale bars in all three panels represent 1 cm. Indicated on the top of the figure are the strains used for all three panels: DK1622 (WT), YZ812 (Δ*stkA*), YZ813 (Δ*stkB*), and YZ910 (Δ*stkC*).

are more important players in S-motility whereas StkC may influence the organization of swarms but not the overall rate of swarming by S motility.

## The *stk* mutants show defects in developmental aggregation consistent with their EPS phenotypes

The *stk* mutants were examined for development under nutrient limitation (Fig. 5C). With respect to both fruiting body morphology and the completeness of aggregation, the Δ*stkA* mutant was the most defective followed by Δ*stkB* and Δ*stkC*. While some of the aggregates of Δ*stkC* are elongated, their distribution and number are the most similar to those of the WT. The aggregates formed by the Δ*stkB* mutant darkened as those of the WT, but they appeared less well organized and more variable in size and number. While the Δ*stkA* mutant showed signs of aggregation in the center of the bacterial lawn, these aggregates are smaller and more numerous. On the edge of the lawn, Δ*stkA* cells appeared to move outward with no signs of aggregation. These results are consistent with the varying degrees of EPS defects of *stk* mutants as Δ*stkA* had the most severe EPS phenotype, followed by Δ*stkB* and Δ*stkC*.
## DISCUSSION & CONCLUSIONS

To summarize, two *stk* insertion mutants were isolated in two separate genetic screens based on their altered EPS phenotypes. Further analysis indicated that both a *stkA* in-frame deletion and an insertion resulted in EPS overproduction in the WT background. While *stkA* mutations suppressed Δ*pilA* in EPS regulation, they failed to restore EPS production to a Δ*difA* mutant. Both *stkB* and *stkC* deletions resulted in varying reductions in EPS production and surface motility, consistent with the correlation between EPS and motility observed previously (*Xu et al., 2005*). These results established that the three genes at the *stk* locus are important for EPS production in *M. xanthus*, albeit to different degrees. The observation that the *stkA* mutant displayed reduced swarming by S motility (Fig. 5B) indicates that optimal S motility requires a fine balance or a tight regulation of EPS production; too little or too much apparently results in reduced efficiency of spreading through S motility (*Xu et al., 2005*).

StkA is a member of the Hsp70 protein family (*Weimer et al., 1998*). The prototype Hsp70 is the *E. coli* chaperone DnaK (*Genevaux, Georgopoulos & Kelley, 2007*). It functions as part of a molecular machine with DnaJ and GrpE, its partners or co-chaperones, to facilitate the folding of nascent polypeptides and the refolding of denatured or misfolded proteins. These proteins are induced by heat shock and confer thermotolerance to *E. coli* once induced. Multiple lines of evidence indicate that StkA is a negative regulator of EPS production in *M. xanthus*. Previously, StkA was not found to be induced by heat shock (*Otani et al., 2001*) and thus not a typical bacterial Hsp70. Instead, our genetic epistasis results here support a model wherein StkA functions as a negative regulator downstream of T4P but upstream of the Dif chemotaxis protein in the EPS regulatory pathway. Previous results left little doubt that StkA is critical for the production of fibrils (*Dana & Shimkets, 1993*), of which EPS is a major constituent (*Behmlander & Dworkin, 1994*). The results here demonstrate that StkA itself is a negative regulator of EPS and lies downstream of T4P and upstream of Dif in the EPS regulatory pathway in *M. xanthus*.

As a component of the EPS signaling pathway, StkA may modulate the function of other EPS regulators in a chaperone-like capacity or it may act directly as a signaling protein in an unknown manner. In this context, it is noted that SglK, another *M. xanthus* Hsp70 homologue, has the opposite function in EPS regulation when compared with StkA (*Weimer et al., 1998*; *Yang, Geng & Shi, 1998*). That is, a *sglK* mutant is EPS⁻ and has no S motility. It is surprising that there are 15 Hsp70-like proteins encoded by the *M. xanthus* genome (*Goldman et al., 2006*). This is in contrast to *E. coli* which codes three Hsp70 members on its genome (*Genevaux, Georgopoulos & Kelley, 2007*). Besides the canonical heat shock protein DnaK, HscA and HscC are the other two Hsp70-like proteins in this enteric bacterium. HscA is a specialized chaperone that facilitates the assembly and maturation of iron-sufur [Fe-S] proteins. HscC appears to be involved in response to more general stress including UV exposure through mechanisms that is not entirely clear. If StkA functions as a chaperone like HscA, it may facilitate a negative regulator of EPS to attain or maturate to its native and active state. If StkA is a signaling protein, it may function in a similar fashion as Ssz1, a regulatory Hsp70 in yeast (*Prunuske et al., 2012*).

The identification of the direct target of StkA will provide insights into the mechanisms of EPS regulation by this member of Hsp70 proteins in *M. xanthus*.

It is unclear whether StkB and StkC function in a regulatory or a biosynthetic capacity. In comparison with the *stkB* and the *stkC* deletions, the *stkB1*::*Tn* mutant has a more severe EPS phenotype (Fig. 1). This suggests that the *stkB1*::*Tn* mutation is polar on *stkC* (Fig. 2) and that StkB and StkC have overlapping or redundant functions. The homology of StkC to PhaE (*Han et al., 2010*) may be taken as circumstantial evident that StkC as well as StkB are EPS biosynthetic enzymes. However, the homology of StkB to NSLTPs (*Lev, 2010*; *Schroeder et al., 2007*) leads to ambiguities on whether there is indeed an overlapping function for these two proteins. NSLTPs are involved not only in lipid metabolism but also in cell signaling (*Lev, 2010*; *Schroeder et al., 2007*), which could mean both function in cell signaling instead of the biosynthetic process. Further investigations are necessary to better understand the roles of StkB and StkC in *M. xanthus* EPS production.

## ACKNOWLEDGEMENTS

RAW was partially supported by the Post Baccalaureate Research and Education Program (PREP) and the Multicultural Academic Opportunities Program (MAOP) at Virginia Tech.

### Funding

This work was partially supported by the National Science Foundation Grant MCB-1417726, the National Institute of Health Grant GM071601, and the Fralin Life Science Institute to Zhaomin Yang. The funders had no role in study design, data collection and analysis, decision to publish, or preparation of the manuscript.

### Grant Disclosures

The following grant information was disclosed by the authors:
National Science Foundation: MCB-1417726.
National Institute of Health: GM071601.
Fralin Life Science Institute.

### Competing Interests

The authors declare there are no competing interests.

### Author Contributions

- Pamela L. Moak conceived and designed the experiments, performed the experiments, analyzed the data, wrote the paper, prepared figures and/or tables, reviewed drafts of the paper.
- Wesley P. Black conceived and designed the experiments, performed the experiments, reviewed drafts of the paper.
- Regina A. Wallace performed the experiments, prepared figures and/or tables, reviewed drafts of the paper.

- Zhuo Li performed the experiments.
- Zhaomin Yang conceived and designed the experiments, analyzed the data, wrote the paper, prepared figures and/or tables, reviewed drafts of the paper.

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
