# Peer review of "The Hsp70-like StkA functions between T4P and Dif signaling proteins as a negative regulator of exopolysaccharide in Myxococcus xanthus"

_PeerJ, doi:10.7717/peerj.747_

## Round 0.1 · original submission · Minor Revisions

Both reviewers have found the underlying experimental approaches addressing your hypotheses, and the conclusions drawn therefrom, to be "solid, significant and straightforward". Incorporating their suggestions, particularly concerning the discussion, should improve communication of the main results and significance of this work. In addition, please provide the supplemental information suggested by reviewer 1, NCBI gene accession numbers for all genes discussed (eg. http://www.ncbi.nlm.nih.gov/gene/?term=txid246197[Organism:noexp] in either Gene_ID or MXAN_#### locus tag format), and clarify how the Calcofluor white fluorescence was measured (qualitatively as implied, or quantitatively). Please also incorporate the minor copy changes to typos or inconsistencies.

Reviewer 1 ·

Basic reporting

Table 1 and elsewhere: The Tn-insertion mutants and in-frame deletion mutants should be better distinguished in this table and the text.
Figures 3 and 4: These figures should be formatted in the same way—it will make comparison easier to the reference wt. Currently, the fonts, scales, bars, and gridlines are all a bit different, which is a bit confusing since they show the same type of data with one sample presumably being the same. Lastly, since these are the quantitative results, it should be stated in the legend that these results are determined by “trypan blue” liquid assay.
The basics of Hsp70 proteins should be introduced. In Line 291, it is explained that stkA is not “a typical bacterial Hsp70”, but a bit more background could be given. Line 313—are there other predicted Hsp70 proteins for M. xanthus?

Experimental design

No concerns. The methods are clearly described.

Validity of the findings

Is there any information to be gained by the fact that the stkA::Tn mutant is towards the C-terminus? Were there differences between this transposon mutant and the clean mutant that were not described?
Line 324: Since supplemental files seem to be allowed by PeerJ—these double mutant results could be shown in a supplemental manner.

Additional comments

Table 1 and elsewhere: The Tn-insertion mutants and in-frame deletion mutants should be better distinguished in this table and the text.

This represents a significant and solid effort by the authors. The study is well-done and the story is clearly laid out. A few editorial changes could make some details just a bit more clear.

Reviewer 2 ·

Basic reporting

The authors appear to have followed the PeerJ policies. The manuscript is generally concise and quite readable. References are in order and the introduction provides sufficient background to give the reader knowledge to understand the new data.

Experimental design

The authors provide an analysis of StkA (a DnaK/HSP70 homolog), StkB (a sterol carrier protein 2 homolog) and StkC (a PhaE homolog). Significantly, mutations in stkA restore production of EPS in a pilA mutant. The next two genes in this apparent operon, stkB and stkC, affect EPS production, but to a lesser extent than stkA. The authors constructed in-frame deletions and a number of double and triple mutants to gain a clear understanding of the function of these genes. They use a standard set of assays that are grounded in the literature and can be reproduced in other laboratories. Experiments were repeated multiple times and are likely to be statistically significant. Error bars are provided.

Validity of the findings

The authors conclude that StkA acts between T4P and Dif pathways to regulate EPS production. StkB and C appear to modulate the regulation. The results are quite solid and straightforward. The conclusions are supported by the data provided. The authors are careful not to over interpret their results.

Additional comments

General comments: It would help to give a NCBI (MXAN) number assignment to these genes. Nothing is given until page 12. What is the phenotype of a stkB pilA or stkC pilA mutant? On page 1, the order of the author’s names was different than on page 2

Specific comments:
Line 7 cell group formation is an awkward term
Line 14 minimally by deletion of stkC
Line 69 change: the T4P functions as physical sensors to T4P function as physical sensors
Line 82 DnaK and HSP70
Line 98 E coli strains were grown and maintained on Luria Bertani (LB) agar plates or in LB liquid medium
line 104 Shi and Zusman used 0.3% agar – is there a reason to use 0.4%.
line 113 This sentence “The site of a Tn insertion in a mutant of interest was identified as has been described” is very cryptic. Can you simply give a useful name?
Line 122 change withr to with
Line 147 eliminate ‘in the manuscript’
Line 160 The Rubin reference is not needed here as a specific transposon is not mentioned.
Line 163 Were only two EPS- mutants found among 20,000? Could you give some numbers here to give the reader a sense of the number of EPS genes?
Line 168 Could be shortened to read: containing the dye Calcofluor white (CW).
Line 169 This sentence is redundant and can be eliminated: This observation confirmed that BY801 harbors a suppressor of ΔpilA mutation in EPS production.
Line 171 16 were examined; were there any EPS negative transformants?

P11 legend line 3: fluorescence. Was the fluorescence quantified?
P11 legend How were these spots photographed? Were these done using a microscope with UV illumination from the top?
P15 Fig 3 EPS overproduction is an interesting phenotype. How does overproduction of EPS affect motility (in a WT background)? Does mixing the stkA mutant with a WT impede gliding of the WT?
Line 225 As a DNA homolog, it seems unlikely that SktA regulates by binding directly to DNA. Any evidence (from in silico analysis perhaps) that it interacts with another protein?
Line 228 likely polar (not polarly); did you look at the levels of transcript to check for polar (increase or decrease) effects?
Line 229 MXAN
Line 249 It would help to describe the actual WT characteristics (pigmentation, opacity, roughness and edges) for the people who don’t work in this field.
Line 251 Is seems surprising that the StkA and B mutants showed poor swarming on hard medium. Was the ratio (Shi and Zusman) different? Did you look at gliding by time-lapse video?
Line 253 What is more vivid in color mean?
Line 256 Is it known that EPS contributes to colony color? This statement seems misleading.
Line 259 The size of the colony is not important – the data reported should be the expansion rate over time.
P19 Fig 5 The YZ910 colony (top right) looks pixelated.
P19 Fig 5 The numbers of heat-resistant spores should be reported.

---

## Round 0.2 · accepted · Accept

I have reviewed all the manuscript changes and believe that your careful attention to and discussion of each point raised by the reviewers improve the communication of your results. The revisions in the second paragraph of the Discussion are an aid to the non-specialist reader and the methodological description ambiguities have been resolved both in the main text and in the figure legends.